# Nickel-catalyzed reductive thiolation and selenylation of unactivated alkyl bromides

Yi Fang[1], Torben Rogge[2], Lutz Ackermann [2], Shun-Yi Wang [1] & Shun-Jun Ji[1]

Chalcogen-containing compounds have received considerable attention because of their manifold applications in agrochemicals, pharmaceuticals, and material science. While many classical methods have been developed for preparing organic sulfides, most of them exploited the transition-metal-catalyzed cross-couplings of aryl halides or pseudo halides with thiols or disulfides, with harsh reaction conditions usually being required. Herein, we present a user-friendly, nickel-catalyzed reductive thiolation of unactivated primary and secondary alkyl bromides with thiosulfonates as reliable thiolation reagents, which are easily prepared and bench-stable. Furthermore, a series of selenides is also prepared in a similar fashion with selenosulfonates as selenolation reagents. This catalytic method offers a facile synthesis of a wide range of unsymmetrical alkyl-aryl or alkyl-alkyl sulfides and selenides under mild conditions with an excellent tolerance of functional groups. Likewise, the use of sensitive and stoichiometric organometallic reagents can be avoided.

[1] Key Laboratory of Organic Synthesis of Jiangsu Province, College of Chemistry, Chemical Engineering and Materials Science & Collaborative Innovation Center of Suzhou Nano Science and Technology, Soochow University, Suzhou 215123, China. [2] Institut für Organische und Biomolekulare Chemie, Georg-August-Universität, Tammannstraße 2, 37077 Goettingen, Germany. Correspondence and requests for materials should be addressed to S.-Y.W. (email: shunyi@suda.edu.cn) or to S.-J.J. (email: shunjun@suda.edu.cn)

The development of novel methods for the construction of sulfides has attracted increasing attention for their key importance in pharmaceuticals, functional materials, and organic syntheses[1–5]. For example, Griseoviridin, first isolated from *Streptomyces griseus*, is a representative member of Streptogramin antibiotics[6]. Viracept is being used as anti-human immunodeficiency virus drug along with other medications[7].

The most classical and important method for the synthesis of alkyl sulfide compounds is the substitution reaction of alkyl halides and mercaptans under strong alkaline reaction conditions. However, these reactions have several drawbacks such as low yields, limited substrate scope, as well as the unpleasant odor of mercaptans. Developing milder and more efficient approaches for the synthesis of organic sulfides continue to be highly desirable. Transition-metal-catalyzed C–S bond cross-coupling reactions have attracted significant attention in recent years[8,9]. The traditional cross-coupling reactions of aryl halides or aryl boronic acids (activated $C_{sp2}$–X) with alkyl thiols have been well developed, although harsh reaction conditions are usually required (Fig. 1a). Noble metals, such as palladium[10–12], rhodium[13–15], gold[16,17], and silver[18], constitute the majority of transition-metal catalysts in $C_{sp2}$–S bond formations. In contrast, the utilization of inexpensive metals, including iron[19,20], copper[21,22], cobalt[23,24], nickel[25–33], and manganese[34,35] is more desirable in the coupling of aryl halides(pseudohaildes) with alkyl thiols. Among the reported methods, thiols and their oxidized derivatives are usually used as thiolation agents. Yet, most thiols are highly toxic compounds and the commercially available alkyl thiols or alkyl disulfides are few, which significantly limit their applications and substrate scope. Lee and co-workers reported a one-pot synthesis of unsymmetrical sulfides using KSAc as the sulfuration agent[36]. Through a similar strategy, Zhou's group achieved the construction of such compounds employing KSCN instead of KSAc[37]. Ma's group[38] and Jiang's group[39] have reported the synthesis of aryl and alkyl sulfides with sulfur powder and $Na_2S_2O_3$ as sulfuration agents, respectively. In addition, methods for the formation of C–S bonds involving electrophilic benzenesulfonothioates were also explored (Fig. 1b). For instance,

organometallic reagents such as Grignard reagents[40,41] and organolithium compounds[42] can react with benzenesulfonothioates to produce unsymmetrical sulfides. However, organometallic reagents are air- and moisture-sensitive and the methods suffer from limited substrate scope and poor chemoselectivity. Recently, transformations of benzenesulfonothioates as sulfuration agents with transition metals have been reported. Thus, Ruijter and co-workers reported a copper-catalyzed three-component synthesis of isothioureas from isocyanides, benzenesulfonothioates, and amines[43]. Xu and co-workers have also reported a copper-catalyzed $C_{sp2}$–S bond formation for the synthesis of 5-thiotriazoles[44].

While the transition-metal-catalyzed cross-coupling reactions of $C_{sp2}$-X have been well developed, the construction of $C_{sp3}$–S bonds with unactivated alkyl halides are more challenging due to undesired β-hydride elimination and homodimerization pathways. Compared to classical cross-coupling processes, the recent years have witnessed the development of nickel-catalyzed reductive coupling reactions[45–50], which have been recognized as powerful and effective tools for converting alkyl halides into useful molecules under mild reaction conditions while avoiding the use of organometallic reagents. Electrophiles, such as $CO_2$[51], isocyanates[52], and acyl chlorides[53] have been successfully employed as coupling partners with alkyl halides. However, to the best of our knowledge, $C_{sp3}$–S bond formations through nickel-catalyzed reductive coupling reactions have not been reported to date. Herein, we present the nickel-catalyzed reductive thiolation of unactivated primary and secondary alkyl bromides with thiosulfonates as reliable thiolation reagents to afford a wide range of unsymmetrical alkyl-aryl or alkyl-alkyl sulfides. Furthermore, a series of selenides was also prepared in a similar fashion with selenosulfonates as selenolation reagents. This protocol is easy to handle, scalable and proceeds smoothly with excellent tolerance of functional groups (Fig. 1c).

## Results

**Reaction condition optimization for synthesis of 3.** The initial investigation was focused on the reaction of 1-bromo-3-

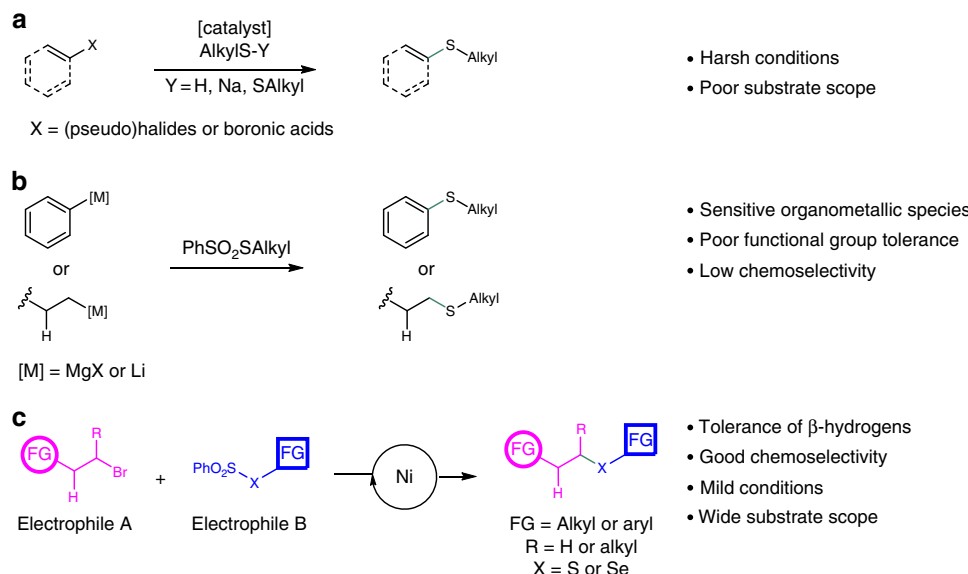

**Fig. 1** Unsymmetrical alkyl-thiolation through different processes. **a** In traditional methods, transition-metal-catalyzed coupling of activated $C_{sp2}$–X with alkyl thiols were well developed, however, harsh reaction conditions are usually required. **b** Compared to traditional thiolation reagents, thiosulfonates are easily prepared and bench-stable, which can react with Grignard reagents or organolithium reagents to generate corresponding alkyl sulfides. **c** Nickel-catalyzed reductive thiolation of unactivated alkyl bromides with thiosulfonates as reliable thiolation reagents was described. This reaction is easy to handle, scalable, and proceeds smoothly with excellent tolerance of functional groups

**Table 1 Optimization of reaction conditions for nickel-catalyzed reductive aryl-thiolation**

| Entry | [Ni] | Ligand | [S] | Solvent | Yield (%)* |
|---|---|---|---|---|---|
| 1 | NiCl₂·glyme | **L5** | **2a** | DMF | 98 |
| 2 | NiBr₂·glyme | **L5** | **2a** | DMF | 97 |
| 3 | NiF₂ | **L5** | **2a** | DMF | 0 |
| 4 | Ni(OAc)₂ | **L5** | **2a** | DMF | Trace |
| 5 | Ni(PPh₃)₂Cl₂ | **L5** | **2a** | DMF | 99 |
| 6 | Ni(PPh₃)₂Cl₂ | **L1** | **2a** | DMF | 93 |
| 7 | Ni(PPh₃)₂Cl₂ | **L2** | **2a** | DMF | >99 |
| 8 | Ni(PPh₃)₂Cl₂ | **L3** | **2a** | DMF | 68 |
| 9 | Ni(PPh₃)₂Cl₂ | **L4** | **2a** | DMF | 84 |
| 10 | Ni(PPh₃)₂Cl₂ | **L6** | **2a** | DMF | 97 |
| 11 | Ni(PPh₃)₂Cl₂ | none | **2a** | DMF | 21 |
| 12 | Ni(PPh₃)₂Cl₂ | **L2** | **2a** | DMA | 92 |
| 13 | Ni(PPh₃)₂Cl₂ | **L2** | **2a** | THF | 0 |
| 14 | Ni(PPh₃)₂Cl₂ | **L2** | **2a** | MeCN | 0 |
| 15 | Ni(PPh₃)₂Cl₂ | **L2** | (PhS)₂ | DMF | 91 |
| 16 | Ni(PPh₃)₂Cl₂ | **L2** | **2a** | DMF | 21† |

Reaction conditions: **1a** (0.2 mmol, 1.0 equiv.); **2a** (0.22 mmol, 1.1 equiv.); [Ni] (5.0 mol%); ligand (7.5 mol%); Mn (0.3 mmol, 1.5 equiv.); DMF (1 mL); N₂ atmosphere; 30 °C; 5 h
*Yields were determined by GC with tridecane as the internal standard
†Zn powder (1.5 equiv.) instead of Mn powder was used

phenylpropane **1a** with *S*-phenyl benzenesulfonothioate **2a** (the general synthesis of the electrophiles is described in the Supplementary Information) as the thiolation reagent. As briefly illustrated in Table 1 (see Supplementary Tables 1-4 for more details), after systematic screening with **L5** as the ligand, Ni(PPh₃)₂Cl₂ was chosen as catalyst because of its cost-effective nature (Table 1, entries 1–5). Next, we conducted the ligand optimization with a series of bipyridine ligands and structurally similar phenanthroline ligands. As a result, ligands lacking *ortho* substituents provided inferior results (Table 1, entries 6–9). In the absence of ligand, the product was observed in only 21% yield (Table 1, entry 11). The utilization of **L2** resulted in high yields, we thus could observe the desired product in nearly quantitative yield (Table 1, entry 7). Other ligands, such as phosphine ligands, could not give improved results (see Supplementary Table 2). After optimizing the reaction medium, it was found that amide solvents were crucial for the successful transformation of **1a**. No desired product could be observed in tetrahydrofuran (THF) or MeCN (Table 1, entries 13 and 14). As shown in entries 15 and 16 (Table 1), the use of other thiolation and reducing reagents resulted in significantly diminished yields of product **3a**.

**Substrate scope of 3.** With the optimized reaction conditions in hand, we explored the substrate scope of the transformation (Table 2). Hence, primary alkyl bromides bearing various functional groups were employed in the reactions with *S*-phenyl benzenesulfonothioate **2a**. To our delight, the desired sulfide compounds could be isolated in good to excellent yields (**3a–n**). A wide range of substituents, including nitrile, ester, chloride, alcohol and alkene, were well-tolerated under these mild reaction conditions. Heteroaromatic substrate and estrone derivative were successfully converted to the corresponding products (**3k, 3l**). Notably, alkyl borate was also tolerated under the standard reaction conditions, illustrating the outstanding chemoselectivity of our approach (**3f**). In addition, alkyl dibromides were also successfully employed to provide the disulfide products, which could be used as potential ligands (**3m, 3n**). Fortunately, a series of alkyl selenides (**3o–r**) were synthesized in good yields with *Se*-phenyl benzenesulfonoselenoate as the selenylation reagent under otherwise identical reaction conditions. Next, we investigated the thiosulfonate scope with a wide range of benzenesulfonothioates bearing different functional groups. Notably, when thiosulfonates with amino group or heteroaromatic substrate were employed, the reactions did not occur smoothly under standard conditions, but higher ligand loading led to satisfactory results (**3w–z**). In view of the importance of essential amino acids, bromoserine was successfully converted to the corresponding cysteine and selenocysteine derivates in good yields (**3aa–ad**).

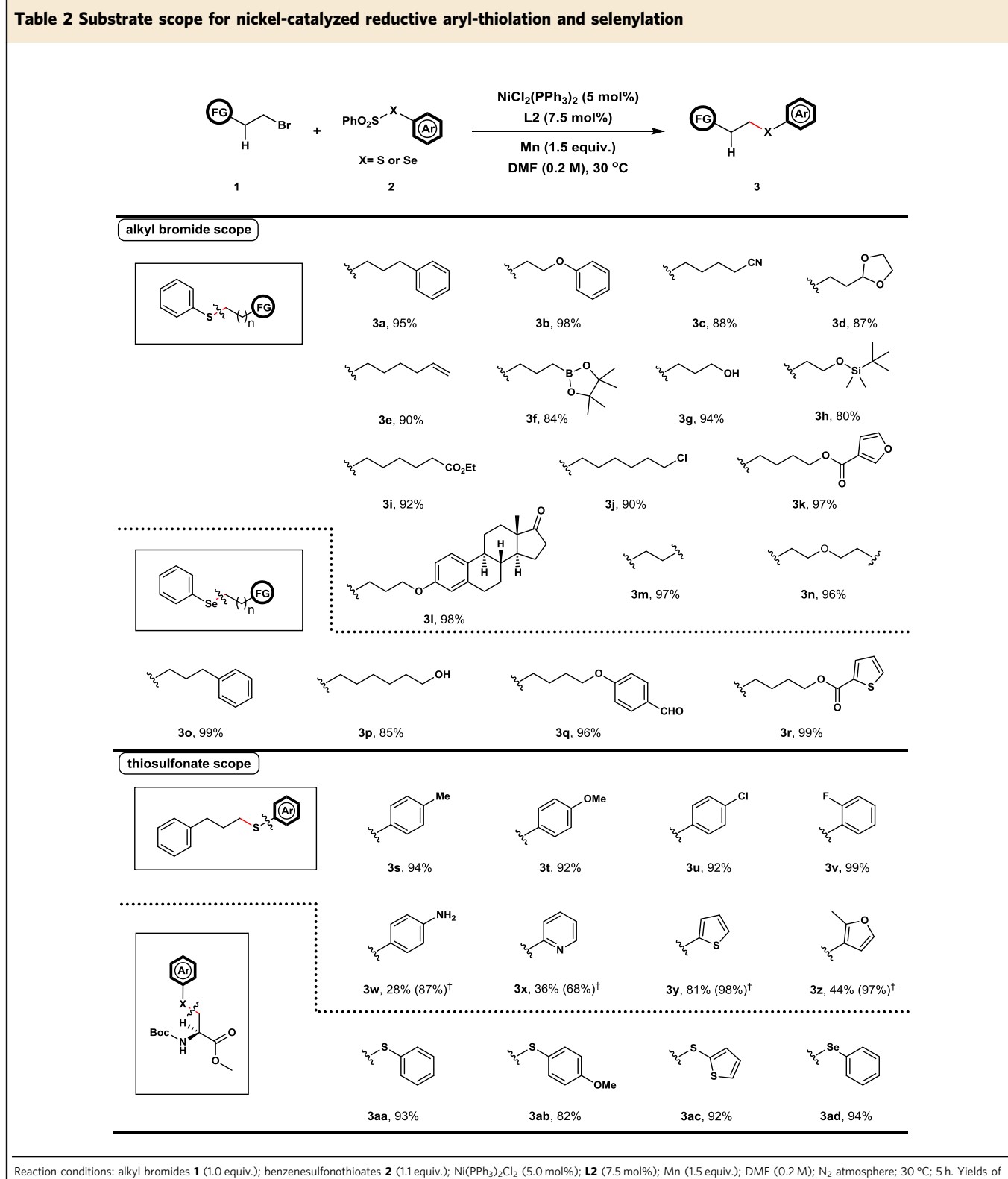

**Table 2 Substrate scope for nickel-catalyzed reductive aryl-thiolation and selenylation**

Reaction conditions: alkyl bromides **1** (1.0 equiv.); benzenesulfonothioates **2** (1.1 equiv.); Ni(PPh₃)₂Cl₂ (5.0 mol%); **L2** (7.5 mol%); Mn (1.5 equiv.); DMF (0.2 M); N₂ atmosphere; 30 °C; 5 h. Yields of isolated products are given
†**L2** (12.5 mol%), 12 h

With the successful synthesis of unsymmetrical alkyl-aryl sulfides and selenides being established, we began to study the construction of more challenging unsymmetrical alkyl-alkyl sulfides and selenides. Similarly, as illustrated in Supplementary Tables 5–9, after extensive screening of nickel catalysts and ligands, a combination of NiBr₂ and **L5** provided best result. To

our delight, after a systematic investigation of the reactions in the solvent mixture of DMF and MeCN, the yield of **5a** could be increased to 99% after 12 h at 100 °C. The utilization of other thiolation or reducing agents resulted in considerably lower yields.

**Table 3 Substrate scope for nickel-catalyzed reductive alkyl-thiolation and selenylation**

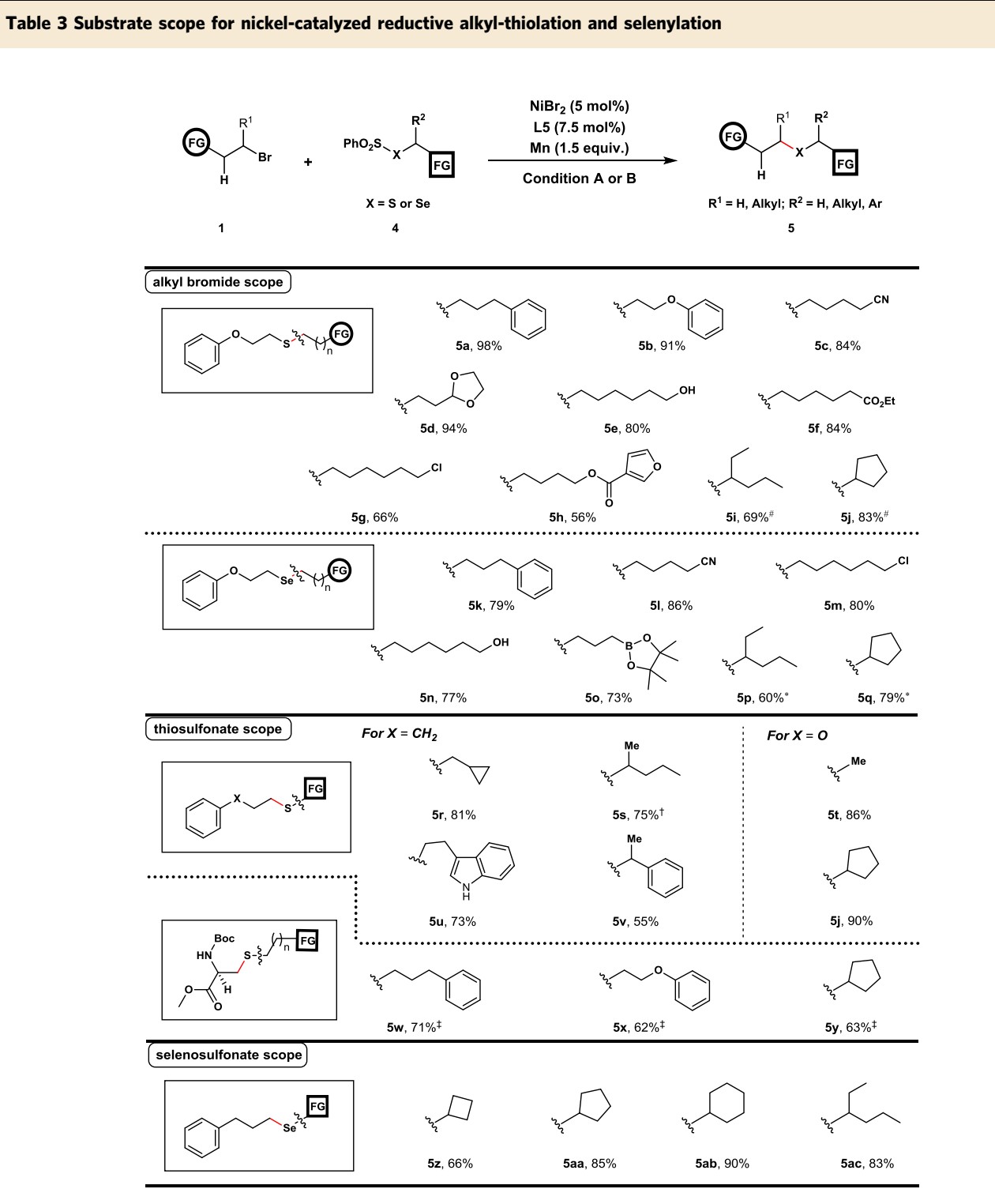

Reaction conditions A: for X = S, alkyl bromides **1** (1.1 equiv.), benzenesulfonothioates **2** (1.0 equiv.), NiBr₂ (5.0 mol%), **L5** (7.5 mol%), Mn (1.5 equiv.), DMF/MeCN (v/v = 2:3, 0.2 M), N₂ atmosphere, 100 °C, 12 h; reaction conditions B: for X = Se, alkyl bromides **1** (1.1 equiv.), benzenesulfonoselenoates **2** (1.0 equiv.), NiBr₂ (5.0 mol%), **L5** (7.5 mol%), Mn (1.5 equiv.), DMF (0.2 M), N₂ atmosphere, 30 °C, 12 h. Yields of isolated products are given
†80 °C, 16 h
‡**1** (0.3 mmol), **2** (0.33 mmol), 80 °C, 16 h
#**1** (3 equiv.) was used, 80 °C
***1** (3 equiv.) was used

**Substrate scope of** 5. Encouraged by these results, we turned our attention to investigate the nickel-catalyzed reductive alkyl-thiolation and selenylation reaction scope with unactivated alkyl bromides (Table 3). Nitrile (**5c**), acetal (**5d**), alcohol (**5e**), and

ester (**5f**) were all compatible in the reductive nickel catalysis manifold. When the alkyl chloride substituent was present, the target product (**5g**) was obtained in 66% yield. To our delight, secondary alkyl bromides were successively transferred to the

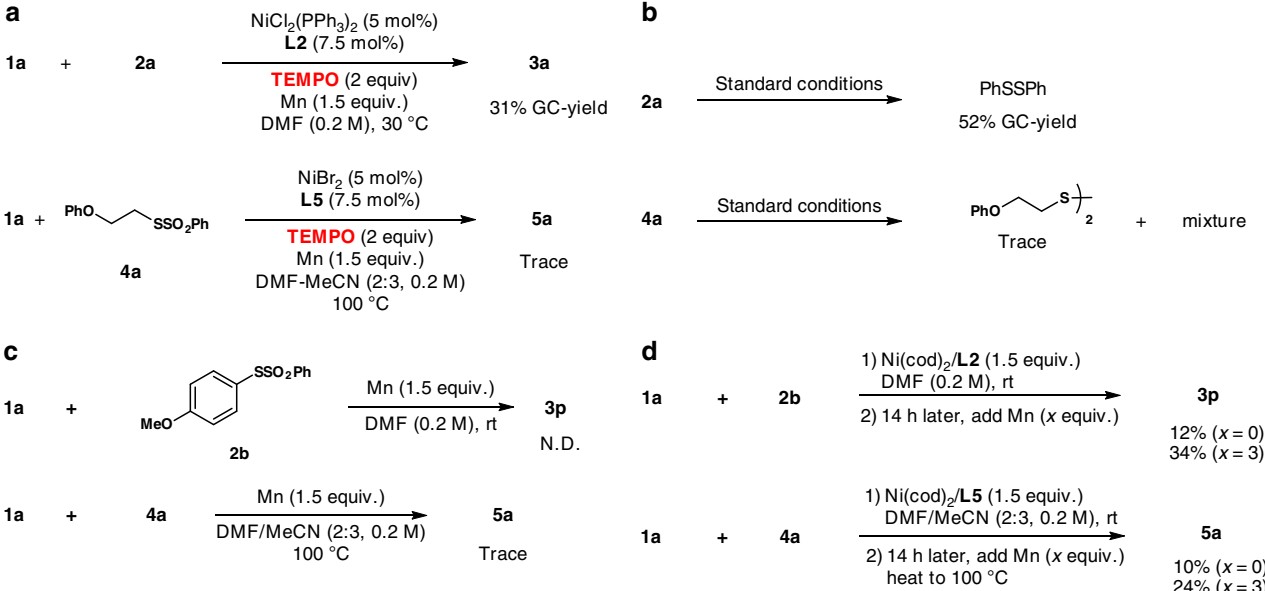

**Fig. 2** Scale-up synthesis and further transformations of sulfides. **a** A cyclic sulfide was generated via intramolecular thiolation. **b** The Ni-catalyst loading could be reduced to 1 mol% without loss of catalytic activity. **c** The alkyl borate could be transferred into different functional groups. **d** Under suitable oxidation conditions, the sulfide could be selectively oxidized to sulfoxide or sulfone

**Fig. 3** Insights into the reaction mechanism. **a** Radical scavenger is added under standard conditions, diminished yields of sulfides are observed. **b** Treatment of **2a** and **4a** under standard reaction conditions leads to the formation of disulfides. **c** Investigation of the role of manganese powder. **d** Stoichiometric experiments

corresponding sulfides in 69% (**5i**) and 83% (**5j**) yield, respectively. Subsequently, we expanded the substrate scope for thiosulfonates. Cyclopropane and indole moieties could be tolerated, and the desired products were obtained with yields of 81% (**5r**) and 73% (**5u**), respectively. It is worth mentioning that we could successfully obtain the methyl sulfide compound (**5t**) with a yield of 86%, which is difficult to synthesize by traditional methods. In addition, the secondary thiosulfonates could also be successfully used in the reaction system (**5j**, **5s**, **5v**). In the same way, this method was applied to the synthesis of cysteine derivatives (**5w–y**). We could not obtain the desired alkyl-alkyl selenides by employing dialkyldiselenides as the selenylation reagents even at 100 °C for 12 h. In contract, the Se-alkyl benzenesulfonoselenoates showed high reactivity and a series of unsymmetrical selenides was synthesized in moderate to good yields using similar conditions at 30 °C (Table 3). Primary alkyl bromides bearing different functional groups were tolerated under the mild conditions. We could obtain the desired selenides in 73–86% yield

(**5k–5o**). Additionally, secondary alkyl bromides were investigated in the reactions. We could obtain the products with yields of 60% (**5p**) and 79% (**5q**), respectively. Next, we investigated the scope of Se-alkyl benzenesulfonoselenoates. Cyclobutyl (**5z**), cyclopentyl (**5aa**), cyclohexyl (**5ab**), and branched (**5ac**) selenides were successfully synthesized in 66–90% yield.

**Further transformation of sulfides.** As described above, sulfides are versatile intermediates and building blocks for the assembly of synthetically useful molecules. With the optimized nickel catalyst in hand, we further applied the alkyl sulfides **3** and **5** in other transformations (Fig. 2). First, **6a** was successfully converted to the thietane derivate (**5ad**) via an intramolecular thiolation (Fig. 2a). Next, a scale-up synthesis of **5a** was achieved with only 1 mol% of nickel catalyst (Fig. 2b). Additionally, we smoothly transformed the product **3f** into **7a** via Suzuki-Miyaura cross-coupling reaction with p-bromonitrobenzene smoothly (Fig. 2c).

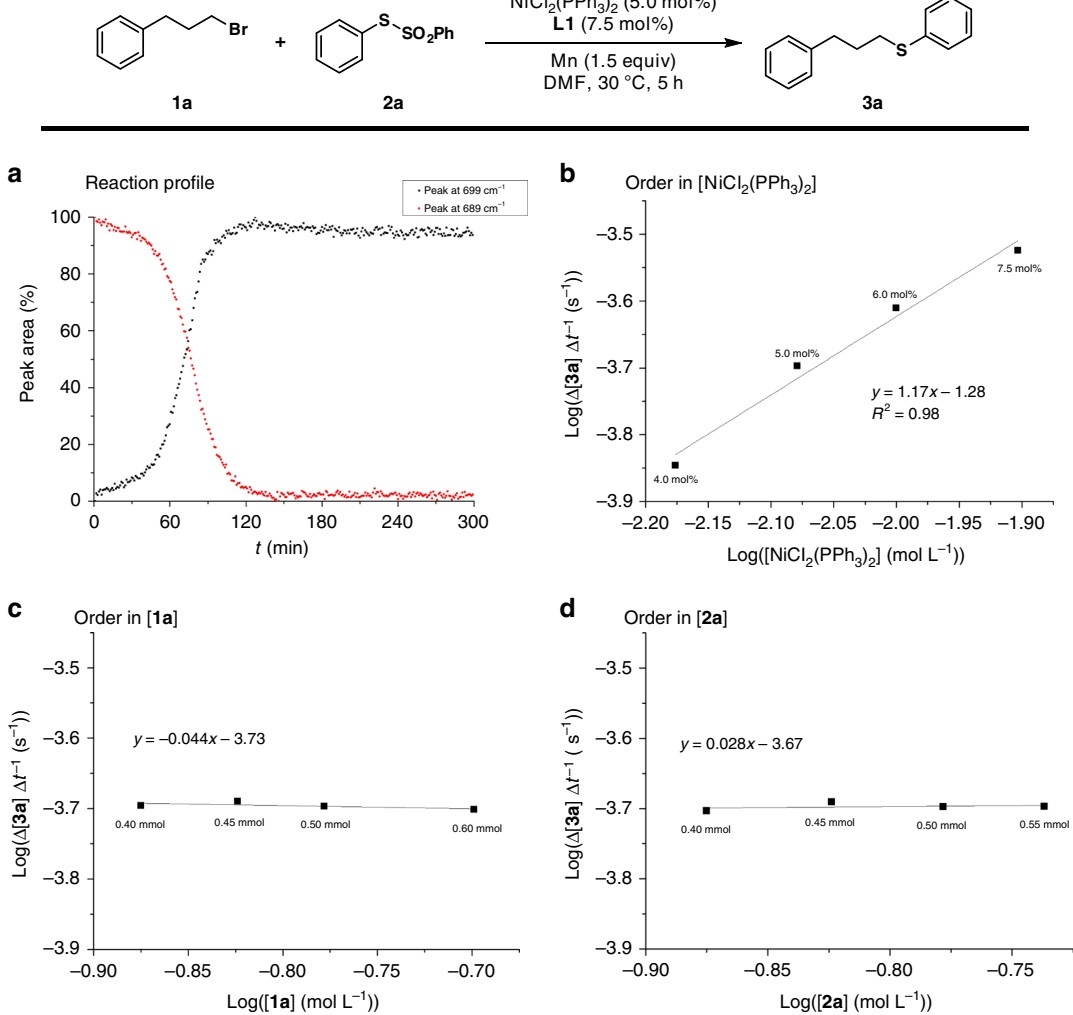

**Fig. 4** Kinetic analysis. **a** Reaction profile. **b** Reaction order in NiCl₂(PPh₃)₂ concentration. **c** Reaction order in alkyl bromide **1a** concentration. **d** Reaction order in benzenesulfonothioate **2a** concentration

Through the treatment of the thus-obtained compound **5a** with different oxidizing agents, we could provide access to sulfoxide **7b** and sulfone **7c** in 83 and 89% yield, respectively (Fig. 2d).

**Mechanistic studies.** Although a comprehensive understanding of the reaction mechanism should await further investigations, preliminary radical-inhibition tests were conducted. Partial inhibition was observed when TEMPO was added. This result indicates that a single-electron-transfer process might occur during the reductive coupling (Fig. 3a). However, the presence of 2,6-di-tert-butyl-4-methylphenol (BHT) only slightly decreased the yields. When treating benzenesulfonothioates **2a** under the standard reaction conditions, the formation of disulfide was observed along with the consumption of **2a** (Fig. 3b). The result shows that an oxidative addition of **2** or **4** to the in situ generated bisligated Ni⁰ intermediate might occur in the initial step. Control experiments were conducted to investigate the role of manganese powder. As shown in Fig. 3c, we could not obtain the desired products with manganese alone, the recovery of benzenesulfonothioates was also possible. Therefore, we could exclude the reduction of benzenesulfonothioates by manganese powder and the formation of alkyl manganese species. More importantly, we found that **3p** and **5a** could be obtained regardless of whether manganese was present or not with stoichiometric amounts of

**Table 4 Radical clock experiment and ratio of linear/cyclic products with respect to catalyst concentration**

| Entry | [NiCl₂(PPh₃)₂] | 3e:3e'[†] |
|-------|----------------|-----------|
| 1 | 5 mol% | 3.5:1 |
| 2 | 7.5 mol % | 9.5:1 |
| 3 | 10 mol% | >30:1 |

[†] Ratios determined by ¹H-NMR of the mixture of **3e** and **3e'**

Ni⁰ catalyst and ligands. Moreover, the addition of manganese powder could increase the yields of products (Fig. 3d).

Detailed kinetic analysis by means of in situ infrared spectroscopy highlighted a first-order dependence with respect to the

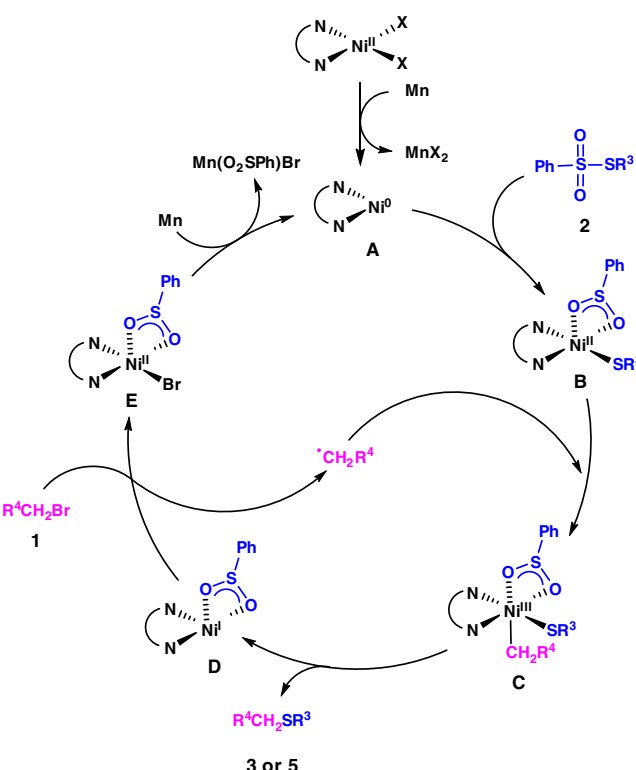

**Fig. 5** Catalytic cycle. Proposed mechanism for the nickel-catalyzed reductive thiolation of unactivated alkyl bromides with thiosulfonates

concentration of NiCl$_2$(PPh$_3$)$_2$ (Fig. 4b). Interestingly, a zeroth-order dependence on the concentration of substrates **1a** and **2a** was revealed, being indicative of a facile C–Br as well as S–S bond cleavage (Fig. 4c, d). These findings are suggestive of the reduction of the nickel catalyst being the rate-determining step.

Reactions with the radical clock precursor **1e** as the substrate resulted in a mixture of the linear and cyclic products **3e** and **3e′**, respectively, providing strong support for the formation of a primary alkyl radical by homolytic C–Br bond cleavage (Table 4). The **3e**:**3e′** ratio was found to increase with higher nickel catalyst concentrations, which is in good agreement with previous reports[54–56], and is indicative of a radical chain mechanism to be operative.

Based on the above experiments and literature reports[55,57,58], a mechanistic scenario was proposed in Fig. 5. The in situ reduction of Ni$^{II}$ by manganese affords Ni$^0$(L)$_2$ (**A**). The oxidative addition of **A** and benzenesulfonothioates **2** furnishes R$^3$S-Ni$^{II}$(L)$_2$ (**B**) and reacts with an alkyl radical to form Ni$^{III}$(L)$_2$ intermediate. The sulfide (**3** or **5**) generates by reductive elimination and a Ni$^I$ (L)$_2$ (**D**) species is formed. The reactive **D** reacts with alkyl bromide **1** to regenerate the alkyl radical and provide **E**. The kinetically relevant further reduction of intermediate **E** regenerates **A**.

## Discussion

In summary, we have developed a nickel-catalyzed reductive chalcogenation of unactivated alkyl bromides with thiosulfonates and selenosulfonates for accessing unsymmetrical sulfides and selenides. The reactions proceed with excellent chemoselectivity under mild reaction conditions, thus tolerating a wide range of functional groups. Additionally, the reaction is amenable to scale-up with low catalyst loading, which enables its applications in

agrochemical and pharmaceutical industries as well as material science. Detailed mechanistic studies provided strong evidence for a facile homolytic C–Br cleavage, and a kinetically relevant nickel reduction.

## Methods

**Synthesis of 3**. In a glovebox, an oven-dried screw-capped 8-mL vial equipped with a magnetic stir bar was charged with NiCl$_2$(PPh$_3$)$_2$ (16.4 mg, 0.025 mmol), 6,6′-dimethyl-2,2′-dipyridyl (**L2**, 6.9 mg, 0.0375 mmol), and Mn powder (41.2 mg, 0.75 mmol). DMF (1 mL) was added via syringe and the mixture was stirred at room temperature for 10 min. The alkyl bromide **A** (0.5 mmol) was added, followed by the addition of aryl-thiosulfonate or selenosulfonate **B** (0.55 mmol) in one portion. Additional DMF (1.5 mL) was subsequently added via syringe. The resulting solution was stirred for 5 h at 30 °C. After this time, the crude reaction mixture was diluted with ethyl acetate (100 mL) and washed with water (20 mL × 3). The organic layer was dried over Na$_2$SO$_4$, filtered, and concentrated. The residue was purified by flash chromatography.

**Synthesis of sulfide 5 with PhSO$_2$SAlkyl**. In a glovebox, an oven-dried screw-capped 8-mL vial equipped with a magnetic stir bar was charged with NiBr$_2$ (3.3 mg, 0.015 mmol), neocuproine (**L5**, 4.6 mg, 0.0225 mmol), and Mn powder (24.8 mg, 0.45 mmol). DMF (0.6 mL) was added via syringe and the mixture was stirred at room temperature for 10 min. The alkyl bromide **A** (0.33 mmol) was added, followed by the addition of alkyl-thiosulfonate **B** (0.3 mmol) in one portion. MeCN (0.9 mL) was subsequently added via syringe. The resulting solution was stirred for 12 h at 100 °C. After this time, the crude reaction mixture was diluted with ethyl acetate (100 mL) and washed with water (20 mL × 3). The organic layer was dried over Na$_2$SO$_4$, filtered, and concentrated. The residue was purified by flash chromatography.

**Synthesis of selenide 5 with PhSO$_2$SeAlkyl**. In a glovebox, an oven-dried screw-capped 8-mL vial equipped with a magnetic stir bar was charged with NiBr$_2$ (3.3 mg, 0.015 mmol), neocuproine (**L5**, 4.6 mg, 0.0225 mmol), and Mn powder (24.8 mg, 0.45 mmol). DMF (0.5 mL) was added via syringe and the mixture was stirred at room temperature for 10 min. The alkyl bromide **A** (0.33 mmol) was added, followed by the addition of alkyl-selenosulfonate **B** (0.3 mmol) in one portion. Additional DMF (1.0 mL) was subsequently added via syringe. The resulting solution was stirred for 12 h at 30 °C. After this time, the crude reaction mixture was diluted with ethyl acetate (100 mL) and washed with water (20 mL × 3). The organic layer was dried over Na$_2$SO$_4$, filtered, and concentrated. The residue was purified by flash chromatography.

**Data availability**. The authors declare that the main data supporting the findings of this study are available within the article and its Supplementary Information files. Additional data are available from the corresponding authors upon request.

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

## Acknowledgements

We gratefully acknowledge the National Natural Science Foundation of China (21772137, 21672157, and 21542015), PAPD, the Major Basic Research Project of the Natural Science Foundation of the Jiangsu Higher Education Institutions (No. 16KJA150002), the Project of Scientific and Technologic Infrastructute of Suzhou (SZS201708), Soochow University, and State and Local Joint Engineering Laboratory for Novel Functional Polymeric Materials for financial support.

## Author contributions

Y.F., L.A., S.Y.W., and J.S.J. conceived and designed the experiments. Y.F. performed the experiments and analyzed the data. T.R. and L.A. performed the detail mechanism studies and analyzed the data. Y.F., L.A., and S.-Y.W. co-wrote the paper. J.S.J. assisted in

editing the manuscript and approved its contents. S.Y.W. directed the project. All authors discussed the experimental results and commented on the manuscript.

## Additional information

**Competing interests:** The authors declare no competing interests.

