## [Peer Review File · Nature Communications]

Reviewer #1 (Remarks to the Author):

In this paper, Wang and Ji described an unprecedented Ni-catalyzed reductive coupling of primary alkyl bromides with thiosulfonates and selenium-modified sulfonates that generated the corresponding disulfides and selenides. This present method tolerates a wider range of functional groups including hydroxyl under mild conditions. After examination of the paper and the SI, I recommend publication of this work provided Major Revisions of the following concerns are addressed.

- 1) In the introduction section, the authors should acknowledge more review papers for the cross-electrophile coupling field. For instance, Weix's *Acc. Chem. Chem.*, Gong's *Top. Curr. Chem.*, Gosmini's *Eur. J. Chem.*, and probably more from 2017.
- 2) Peng's work on Ni-catalyzed reductive coupling of aryl-SH with ArX (*Org. Lett.* 2013, 15, 550) should be cited.
- 3) The scope of alkyl halides is broad for primary alkyl bromides. However, none of the secondary alkyl electrophiles were discussed. In my view, the secondary halides often display similar reactivities to their primary counterparts. Therefore, a more detailed discussion on these class of substrates should be addressed. The quality of this paper may otherwise be eroded without these studies.
- 4) The reaction mechanism is not quite clear. Since nucleophilic addition of aryl thiols to primary alkyl bromides are a highly efficient process under suitable basic conditions, careful studies of the reaction mechanism would be considerably important. The sulfur radical may be further reduced to sulfur anions, therefore, a control study by addition of Ar'SH or alkyl-SH to the reaction conditions should be performed, wherein a different Ar' or alkyl is used in order to determine cross-sulfide product profiles.
- 5) Deuterated alkyl substrates elucidated in Martin paper (*J. Am. Chem. Soc.* 2016, 138, 7504–7507) or Gong's paper (*JACS* 2014, 17645) should be used to determine whether an alkyl radical is involved.
- 6) Also, the authors should study a dependence of the concentration of Ni catalyst loading on the intramolecular liner/cyclization product ratios using 6-bromo-1-alkene as the substrate (see: 2013, 135, 16192–16197).
- 7) The flowing reference should be cited, *ACS Catal.* 2014, 4, 2941–2950. It includes Ni(I) to Ni(III) thiolation of aryl halides.

Reviewer #2 (Remarks to the Author):

The manuscript by Ji and Wang deals with thiolation of alkyl bromides without the use of converting the alkyl halide into a more active species such as a Grignard reagent etc. The sulfonothioate seems like an extremely active coupling partner in this process and highlights the impact and novelty of the paper. The coupling is high yielding and the reductive conditions seem mild at 30°C and have a good substrate scope. Please make sure the reader is aware the mild conditions are not used throughout the study. The paper should be accepted with minor revisions, addition of references and some more experiments to understand the processes involved in the catalytic cycle.

Line 47 and 48: The two sentences (below) here need clarification to specify the authors are referencing aryl halide(pseudohalide) alkyl thiol coupling. "Noble metals, such as palladium¹¹⁻¹³, rhodium¹⁴⁻¹⁶, gold¹⁷⁻¹⁹ and silver²⁰, constitute the majority of transition-metal catalysts in C-S bond formations. In contrast, the utilization of inexpensive metals including iron^{21,22}, copper^{23,24}, cobalt^{25,26}, nickel^{27,28} and manganese^{29,30} are more."

In the case of nickel, the topic of discussion the references need to be more thorough to include papers: Jouffroy, M.; Kelly, C. B.; Molander, G. A. *Org. Lett.* 2016, 18, 876. Vara, B. A.; Li, X.; Berritt, S.; Walters, C. R.; Petersson, E. J.; Molander, G. A. *Chem. Sci.*, 2018,9, 336-344; Oderinde, M. S.; Frenette, M.; Robbins, D. W.; Aquila, B.; Johannes, J. W. *J. Am. Chem. Soc.* 2016, 138, 1760; K D. Jones, D. J. Power, D. Bierer, K M. Gericke, S G. Stewart, *Org. Lett.*, 2018, 20, 208–211, Pei Guan a, Changsheng Cao, Yun Liu a, Yunfei Li, Pan He, Qian Chen, Gang Liu, Yanhui Shi, *Tetrahedron Letters* 53 (2012) 5987–5992. and for decarbonylative method later in the text Naoko Ichiishi, Christian A. Malapit, Łukasz Woźniak, Melanie S. Sanford *Org. Lett.* 2018, 20, 44–47.

Line 43 and 54: The odour issue does not need to be mentioned twice.

Line 95 and 96: "showed an excellent outcome" should be "resulted in high yields"

The general synthesis of the electrophiles needs to be mentioned in the text as well, or as a reference. The synthesis of S-(pentan-2-yl) benzenesulfonothioate and S-(3-phenylpropyl) benzenesulfonothioate is moderate yielding and some of the novelty this coupling is lost. Can these be improved during this review.

Line 115: alkene, were well tolerated under these mild reaction conditions. Heteroaromatic substrate

Line 141: Although later couplings are carried out at 100°C (Table 3), these changes in conditions specifically the temperature are not mentioned in the text.

Line 142: The utilization of other thiolation or reducing agent(s) resulted in considerably lower yields

I think it is necessary to include a more detailed mechanistic study. The authors eliminate a few probable coupling pathways and then propose an intermediate of RS-Ni(L)₂. Some more evidence for a Ni(I) species (or kinetic studies) here would be nice to conclude the paper. Plus the inclusion of a proposed catalytic cycle would be more thorough.

Reviewer #3 (Remarks to the Author):

This manuscript describes nickel-catalyzed reductive thiolation and selenylation of unactivated alkyl bromides. The concept is novel that Csp³-S bond formations of unactivated primary alkyl bromides by using nickel-catalyzed reductive coupling reaction. They also afforded not only a wide range of unsymmetrical alkyl-aryl but also alkyl-alkyl sulfides under mild reaction condition. In addition, this is the first report for the coupling reaction of alkyl halide and sulfide to construct C-S bond formations.

This manuscript was recommended for publication in Nature Communication when the following comments are addressed.

1. In an unsymmetrical alkyl-aryl sulfide substrate scope, the example of selenylation has only five (four for alkyl bromide scope and one for thiosulfonate scope). Moreover, in an unsymmetrical alkyl-alkyl sulfide substrate scope, there was no example of selenylation. The example for thiolation is great, but if the author mention about selenylation as in the title, it is necessary to show some examples of selenylation in Table 3 as well. (Table 2. and Table 3)
2. In the TEMPO experiment, the yield of product (3a, 5a) is relatively low and trace, but no TEMPO adduct appears at all, it seems to lack information to be proposed SET process. For example, additional data such as a radical clock experiment, a radical trapping, or a Stern-Volmer quenching experiment may be needed. (Figure 3-a)
3. In figure 3, the structure of 4a has only the structure expected in supporting information but not 4a, and the structure of 4a cannot be found in the manuscript. (Figure 3)
4. If the reason for using Ni(COD)₂ in figure 3-d is the same condition for the both reactions, it is necessary to add using Ni(COD)₂ data in the nickel catalyst screening of supporting information table 1. (Figure 3-d, and SI Table 1)
5. Mechanism study should not be mentioned only in manuscript, but addition of scheme is also necessary.

Is the proposed mechanism between 'alkyl-aryl sulfide' and 'alkyl-alkyl sulfide' the same?

And is the proposed mechanism between 'thiolation' and 'selenylation' the same? If that mechanism is the same, add more examples of selenylation scope. If that mechanism is different, it is necessary to explain why there are few examples of selenylation. (Line 196-200 in the manuscript)

6. At line 94, there was mentioned that the absence of a substituent at the ortho-position of the ligand would reduce the yield of the product (L3: 68%, L4: 84%), so how can L1 (91%) be explained? (Line 94, and Table 1)

In addition, there are many typos.

1. Line 55~56: Lee and co-workers reported a "two-step" synthesis of unsymmetrical sulfides -> Lee and co-workers reported a "one-pot" synthesis of unsymmetrical sulfides

2. Table 1: What does "B-1" stand for in the section [S]?

3. Line 145: scope of unactivated bromides (Table 4) -> scope of unactivated bromides (Table 3) :

4. Figure 2, b. compound "3a" has to be corrected to "5a"

The followings has to be corrected ;

Line 184: (Fig. 2a) -> (Fig. 3a), Line 187: (Fig. 2b) -> (Fig 3b), Line 190: (Fig.2c) -> (Fig 3c), Line 196: (Fig. 2d) -> (Fig. 3d) :

Line 193: we found that "3p" and 5a -> we found that "3t" and 5a

Figure 3: c) in the first scheme, compound "3p" has to be corrected to "3t" d) in the first scheme, compound "3p" has to be corrected to "3t"

[Reference Correction]

18. Georgy, M., Boucard, V., Debleds, O., Zotto, C. D. & Campagne, J. M. Gold(III)-catalyzed direct nucleophilic substitution of propargylic alcohols. *Tetrahedron* 65, 1758-1766 (2009).

-> This reference mentioned about gold-catalyst but not exactly about C-S bond formation because there are only three examples. : I do not agree it is the correct reference.

Dear Dr. Giovanni Bottari,

Thanks very much for your kind help and give us a chance to resubmit our work (NCOMMS-18-01630)! According to your suggestion and the referees' comments, we have revised the manuscript. The corrections are listed below.

Response to the referees' comments:

Reviewer #1

- 1) In the introduction section, the authors should acknowledge more review papers for the cross-electrophile coupling field. For instance, Weix's *Acc. Chem. Chem.*, Gong's *Top. Curr. Chem.* Gosmini's *Eur. J. Chem.*, and probably more from 2017.
 - We have added the review papers in the introduction section.
- 2) Peng's work on Ni-catalyzed reductive coupling of aryl-SH with ArX (*Org. Lett.* 2013, 15, 550) should be cited.
 - Peng's (*Org. Lett.* 2013, 15, 550) work has been cited.
- 3) The scope of alkyl halides is broad for primary alkyl bromides. However, none of the secondary alkyl electrophiles were discussed. In my view, the secondary halides often display similar reactivities to their primary counterparts. Therefore, a more detailed discussion on these class of substrates should be addressed. The quality of this paper may otherwise be eroded without these studies.
 - We have examined the reactions of secondary alkyl bromides under the optimized conditions. As expected, the secondary bromides could be successively employed in the construction of sulfides and selenides, and the results have been added in the paper.
- 4) The reaction mechanism is not quite clear. Since nucleophilic addition of aryl thiols to primary alkyl bromides are a highly efficient process under suitable basic conditions, careful studies of the reaction mechanism would be considerably important. The sulfur radical may be further reduced to sulfur anions, therefore, a control study by addition of Ar'SH or alkyl-SH to the reaction conditions should be performed, wherein a different Ar' or alkyl is used in order to determine cross-sulfide product profiles.
 - We have conducted the following experiments, a mixture of sulfides (**4** and **5**) were observed by GC (eq. 1). When 4-methylbenzenethiol **3** was reacted with the alkyl bromide under standard conditions, **4** was obtained (eq. 2). When we mixed **2** and **3** in DMF at rt, the unsymmetric disulfide **6** could be obtained, which could be transferred to **4** and **5** under standard conditions (eq. 3).

- 5) Deuterated alkyl substrates elucidated in Martin paper (J. Am. Chem. Soc. 2016, 138, 7504–7507) or Gong's paper (JACS 2014, 136, 17645) should be used to determine whether an alkyl radical is involved.
 - We have conducted the radical studies by using 6-bromohex-1-ene and the results have been added in the paper and SI.
- 6) Also, the authors should study a dependence of the concentration of Ni catalyst loading on the intramolecular linear/cyclization product ratios using 6-bromo-1-alkene as the substrate (see: 2013, 135, 16192–16197).
 - We have conducted the study and the results have been added in the paper and SI.
- 7) The following reference should be cited, ACS Catal. 2014, 4, 2941–2950. It includes Ni(I) to Ni(III) thiolation of aryl halides.
 - We have cited the reference in the paper.

Reviewer #2

- 1) Line 47 and 48: The two sentences (below) here need clarification to specify the authors are referencing aryl halide(pseudohalide) alkyl thiol coupling. "Noble metals, such as palladium¹¹⁻¹³, rhodium¹⁴⁻¹⁶, gold¹⁷⁻¹⁹ and silver²⁰, constitute the majority of transition-metal catalysts in C-S bond formations. In contrast, the utilization of inexpensive metals including iron^{21,22}, copper^{23,24}, cobalt^{25,26}, nickel^{27,28} and manganese^{29,30} are more."
 - We have corrected the sentences as follow: "Noble metals, such as palladium¹⁰⁻¹², rhodium¹³⁻¹⁵, gold¹⁶⁻¹⁷ and silver¹⁸, constitute the majority of transition-metal catalysts in C_{sp2}-S bond formations. In contrast, the utilization of inexpensive metals including iron^{19,20}, copper^{21,22}, cobalt^{23,24}, nickel²⁵⁻³³ and manganese^{34,35} are more desirable in the coupling of aryl halides(pseudohalides) with alkyl thiols."
- 2) In the case of nickel, the topic of discussion the references need to be more thorough to include papers: Jouffroy, M.; Kelly, C. B.; Molander, G. A. Org. Lett. 2016, 18, 876. Vara, B. A.; Li, X.; Berritt, S.; Walters, C. R.; Petersson, E. J.; Molander, G. A. Chem. Sci., 2018,9, 336-344; Oderinde, M. S.; Frenette, M.; Robbins, D. W.; Aquila, B.; Johannes, J. W. J. Am. Chem. Soc. 2016, 138, 1760; K D. Jones, D. J. Power, D. Bierer, K M. Gericke, S G. Stewart, Org. Lett., 2018, 20, 208–211, Pei Guan a, Changsheng Cao, Yun Liu a, Yunfei Li, Pan He, Qian Chen, Gang Liu, Yanhui Shi, Tetrahedron Letters 53 (2012) 5987–5992. and for decarbonylative method later in the text Naoko Ichiishi, Christian A. Malapit, Łukasz Woźniak,, Melanie S. Sanford Org. Lett. 2018, 20, 44–47.
 - We have added the above references in the paper
- 3) Line 43 and 54: The odour issue does not need to be mentioned twice.
 - We have deleted the odour issue of Line 54.
- 4) Line 95 and 96: "showed an excellent outcome" should be "resulted in high yields"

- “showed an excellent outcome” has been changed to “resulted in high yields”.
- 5) The general synthesis of the electrophiles needs to be mentioned in the text as well, or as a reference. The synthesis of S-(pentan-2-yl) benzenesulfonothioate and S-(3-phenylpropyl) benzenesulfonothioate is moderate yielding and some of the novelty this coupling is lost. Can these be improved during this review.
 - We have added the general synthesis of the electrophiles as a reference [54] in the paper. The yields of benzenesulfonothioate were relatively lower when unactivated secondary alkyl bromides were used. Unfortunately, it is difficult to improve the yields of these benzenesulfonothioate at this moment.
 - 6) Line 115: alkene, were well tolerated under these mild reaction conditions. Heteroaromatic substrate
 - We have corrected the sentence in the paper.
 - 7) Line 141: Although later couplings are carried out at 100°C (Table 3), these changes in conditions specifically the temperature are not mentioned in the text.
 - We have changed the sentence as follow: “To our delight, after a systematic investigation of the reactions in the solvent mixture of DMF and MeCN, the yield of **5a** could be increased to 99% after 12h at 100 °C.”
 - 8) Line 142: The utilization of other thiolation or reducing agent(s) resulted in considerably lower yields
 - We have corrected the sentence in the paper.
 - 9) I think it is necessary to include a more detailed mechanistic study. The authors eliminate a few probable coupling pathways and then propose an intermediate of RS-Ni(L2). Some more evidence for a Ni(I) species (or kinetic studies) here would be nice to conclude the paper. Plus the inclusion of a proposed catalytic cycle would be more thorough.
 - We have conducted the kinetic studies and the results have been added in the paper and SI.

Reviewer #3

- 1) In an unsymmetrical alkyl-aryl sulfide substrate scope, the example of selenylation has only five (four for alkyl bromide scope and one for thiosulfonate scope). Moreover, in an unsymmetrical alkyl-alkyl sulfide substrate scope, there was no example of selenylation. The example for thiolation is great, but if the author mention about selenylation as in the title, it is necessary to show some examples of selenylation in Table 3 as well. (Table 2. and Table 3)
 - We have expanded the scope of selenylation and the examples have been added in Table 3.
- 2) In the TEMPO experiment, the yield of product (3a, 5a) is relatively low and trace, but no TEMPO adduct appears at all, it seems to lack information to be proposed SET process. For example, additional data such as a radical clock experiment, a radical trapping, or a Stern-Volmer quenching experiment may be needed. (Figure 3-a)
 - We have conducted the radical clock experiment with 6-bromohex-1-ene. The ratio of

linear/cyclic products was found to increase with higher nickel catalyst concentrations, which is indicative of a radical chain mechanism to be operative

- 3) In figure 3, the structure of 4a has only the structure expected in supporting information but not 4a, and the structure of 4a cannot be found in the manuscript. (Figure 3)
 - We have added the structure of **4a** in figure 3, and the labels were corrected as well in the SI.
- 4) If the reason for using Ni(COD)₂ in figure 3-d is the same condition for the both reactions, it is necessary to add using Ni(COD)₂ data in the nickel catalyst screening of supporting information table 1. (Figure 3-d, and SI Table 1)
 - The data of using Ni(COD)₂ has been added in Supplementary Table 1.
 - The reason for using stoichiometric Ni(COD)₂ is to examine the role of Ni(0) in the initial step of the reactions. The yields of products were improved when the Mn powder was added, which showed the possibility of reducing of RS-Ni(II) to RS-Ni(I).
- 5) Mechanism study should not be mentioned only in manuscript, but addition of scheme is also necessary. Is the proposed mechanism between 'alkyl-aryl sulfide' and 'alkyl-alkyl sulfide' the same? And is the proposed mechanism between 'thiolation' and 'selenylation' the same? If that mechanism is the same, add more examples of selenylation scope. If that mechanism is different, it is necessary to explain why there are few examples of selenylation. (Line 196-200 in the manuscript)
 - The reactions temperature for 'alkyl-aryl sulfide' and 'alkyl-alkyl sulfide' are different for their different reaction activities. Based on the control experiments and kinetic studies, we think that the proposed mechanism between 'alkyl-aryl sulfide' and 'alkyl-alkyl sulfide' are same. In addition, more selenylation data have been added in the manuscript.
- 6) At line 94, there was mentioned that the absence of a substituent at the ortho-position of the ligand would reduce the yield of the product (L3: 68%, L4: 84%), so how can L1 (91%) be explained? (Line 94, and Table 1)
 - The above results were obtained based on ligands with the same skeleton. Ortho substituted L2 resulted in higher yield than L1 and ortho substituted L5 resulted in higher yield than L4.

We believe that your suggestion and the referees' comments improve the quality of our manuscript. We would like this work to be considered for publication as an article in the **Nature Communications**.

I will be happy to provide if you require any further details. Kindly communicate the result as early as possible and thanking you for the same.

Thank you very much!

Best regards

Sincerely yours

Shun-Yi Wang

Reviewer #1 (Remarks to the Author):

I am satisfied with the authors' responses to the questions I raised for the previous manuscript. My only suggestion for the authors is to reexamine the proposed catalytic cycle. If a radical-chain scenario is considered, the alkyl radical will escape to a bulk solution to find a Ni-SR species, possibly Ln-Ni(II)-SR, see Weix 2013 JACS paper (ref. 56) for details. The current Ni(III) formation in Figure 5 is a cage-rebound model. I suggest publication of this piece of work after such a minor issue being addressed.

Reviewer #3 (Remarks to the Author):

The revised manuscript by Ji addressed all issues which the reviewers asked. Therefore, I agree that this revised manuscript to be published in Nature Communications.

Dear Dr. Giovanni Bottari,

We have revised the manuscript according to the reviewer's suggestion and the response is listed below.

Reviewer #1 (Remarks to the Author):

I am satisfied with the authors' responses to the questions I raised for the previous manuscript. My only suggestion for the authors is to reexamine the proposed catalytic cycle. If a radical-chain scenario is considered, the alkyl radical will escape to a bulk solution to find a Ni-SR species, possibly Ln-Ni(II)-SR, see Weix 2013 JACS paper (ref. 56) for details. The current Ni(III) formation in Figure 5 is a cage-rebound model. I suggest publication of this piece of work after such a minor issue being addressed.

- According to Weix's paper (ref. 56), if a radical chain was operative, then we would expect that the un-rearranged/rearranged ratio would increase at higher catalyst concentrations. The **3e:3e'** ratio was found to increase with higher nickel catalyst concentrations, which was in agreement with a radical-chain scenario. We have modified the catalytic cycle properly. The following is the revised proposed mechanism "Based on the above experiments and literature reports^{56,58,59}, a mechanistic scenario was proposed in Fig. 5. The *in situ* reduction of Ni^{II} by manganese affords Ni⁰(L)₂ (**A**). The oxidative addition of **A** and benzenesulfonothioates **2** furnishes R³S-Ni^{II}(L)₂ (**B**) and reacts with an alkyl radical to form Ni^{III}(L)₂ intermediate. The sulfide (**3** or **5**) generates by reductive elimination and a Ni^I(L)₂ (**D**) species is formed. The reactive **D** reacts with alkyl bromide **1** to regenerate the alkyl radical and provide **E**. The kinetically relevant further reduction of intermediate **E** regenerates **A**."

We hope this revision fulfill the requirements for the publication of our study in Nature Communications.

Thank you very much!

Best regards,

Shun-Yi Wang